

# Improving course evaluation processes in higher education institutions: a modular system approach

İlker Kocaoğlu[1] and Erinç Karataş[2]

[1] Management Information Systems, Baskent University, Ankara, Turkey
[2] Informatics, Ankara University, Ankara, Turkey

## ABSTRACT

Course and instructor evaluations (CIE) are essential tools for assessing educational quality in higher education. However, traditional CIE systems often suffer from inconsistencies between structured responses and open-ended feedback, leading to unreliable insights and increased administrative workload. This study suggests a modular system to address these challenges, leveraging sentiment analysis and inconsistency detection to enhance the reliability and efficiency of CIE processes.
**Background:** Improving the reliability of CIE data is crucial for informed decision-making in higher education. Existing methods fail to address discrepancies between numerical scores and textual feedback, resulting in misleading evaluations. This study proposes a system to identify and exclude inconsistent data, providing more reliable insights.
**Methods:** Using the Design Science Research Methodology (DSRM), a system architecture was developed with five modules: data collection, preprocessing, sentiment analysis, inconsistency detection, and reporting. A dataset of 13,651 anonymized Turkish CIE records was used to train and evaluate machine learning algorithms, including support vector machines, naive Bayes, random forest, decision trees, K-nearest neighbors, and OpenAI's GPT-4 Turbo Preview model. Sentiment analysis results from open-ended responses were compared with structured responses to identify inconsistencies.
**Results:** The GPT-4 Turbo Preview model outperformed traditional algorithms, achieving 85% accuracy, 88% precision, and 95% recall. Analysis of a prototype system applied to 431 CIEs identified a 37% inconsistency rate. By excluding inconsistent data, the system generated reliable reports with actionable insights for course and instructor performance. The purpose of this study is to design and evaluate a new system using the Design Science Research (DSR) approach to enhance the accuracy and reliability of course evaluation processes employed in higher education institutions. The modular system effectively addresses inconsistencies in CIE processes, offering a scalable and adaptable solution for higher education institutions. By integrating advanced machine learning techniques, the system enhances the accuracy and reliability of evaluation reports, supporting data-driven decision-making. Future work will focus on refining sentiment analysis for neutral comments and broadening the system's applicability to diverse educational contexts. This innovative approach represents a significant advancement in leveraging technology to improve educational quality.

Corresponding author
İlker Kocaoğlu,
ilkerkocaoglu1881@gmail.com

# INTRODUCTION

In higher education institutions, course and instructor evaluation (CIE) processes play a significant role in enhancing the quality of education. In these processes, students are generally asked to evaluate both the course and the instructor at the end of the semester. The results of these evaluations provide important feedback to institutional administrators and instructors regarding educational processes. Analyzing this data helps administrators make more accurate, data-driven decisions (*McAfee & Brynjolfsson, 2012*). Current CIE processes typically consist of two sections: the first section includes structured (Likert-type) questions, and the second section comprises an open-ended question where students can write their opinions and suggestions.

In existing CIE implementations, higher education institutions collect the responses from students online and forward them to the relevant administrators without any further processing. However, significant discrepancies can sometimes be observed between the remarks expressed in the open-ended section and the responses given to the structured questions, adversely affecting the accuracy of the evaluation process. In addition, these discrepancies may result in data that does not fully reflect reality and lead to misleading outcomes. Moreover, manual evaluation of this process by administrators causes a loss of time.

This research aims to address the problems identified in current CIE processes, increase their accuracy and reliability, and save administrators time. Employing supervised machine learning and artificial intelligence algorithms, it seeks to enhance the effectiveness of course evaluation systems through sentiment analysis and inconsistency detection.

In the proposed system, an appropriate algorithm is selected to perform a sentiment analysis on the responses given to the open-ended question and determine the students' emotional expressions (positive/negative/neutral). Afterwards, these emotional expressions are compared with the students' responses to the structured questions to detect any inconsistencies. The identified inconsistent data is then excluded from the dataset to be evaluated, thereby producing a course and instructor evaluation report based on reliable data. This study not only aims to improve the existing course evaluation processes but also aspires to introduce a new approach to the effective use of big data analytics and modern technologies in higher education institutions. Consequently, the resulting reports, prepared with objective, consistent, and reliable data, are expected to contribute to the improvement of educational processes.

The reports generated by the developed system provide both administrators and instructors with meaningful insights into the strengths and weaknesses of their courses, thereby contributing to strategic decision-making processes. In this context, the study helps advance a data-driven decision-making culture aimed at improving educational processes.

The "Literature Review" section of the study presents a literature review, while the "Materials and Methods" section offers a detailed explanation of the research methodology and the system architecture used. The "Materials and Methods" section also focuses on the analysis and design processes, along with the development of a prototype application. In the "Conclusion and Discussion" section, the prototype system is implemented and the outcomes are discussed, followed by a general evaluation that highlights the contributions of this study to educational processes.

## LITERATURE REVIEW

Sentiment analysis is a branch of natural language processing that systematically identifies and classifies the emotions present in textual data (*Pang, Lee & Vaithyanathan, 2002*). Over the years, it has been conducted for a variety of purposes (*Grimalt-Álvaro & Usart, 2024*; *Devika, Sunitha & Ganesh, 2016*; *Gonçalves et al., 2013*; *Jiménez et al., 2021*; *Kauffmann et al., 2019*; *Boiy & Moens, 2009*; *Almosawi & Mahmood, 2022*; *Fauziah, Yuwono & Aribowo, 2021*). In the literature, these studies are classified according to their objectives, the tools used for sentiment analysis, and the platforms on which they are performed (*Baragash & Aldowah, 2021*).

Many sentiment analysis studies have focused on student evaluation data in higher education institutions. In *Nasim, Rajput & Haider (2017)*, the authors combined machine learning and lexicon-based approaches to identify the sentiments in student feedback. In *Sivakumar & Reddy (2017)*, the sentiment analysis of course feedback shared by students on social media platforms was conducted to categorize sentiments for the purpose of improving educational processes. *Toçoğlu & Onan (2021)* demonstrated that machine learning-based sentiment analysis methods are effective in student evaluations for higher education institutions, particularly illustrating that using the term frequency-inverse document frequency (TF-IDF) scheme with larger feature sets enhances predictive performance.

In 2022, *Lazrig & Humpherys (2022)* performed machine learning-based sentiment analysis to evaluate students' learning experiences. The study achieved 98% accuracy in predicting positive and negative sentiments, yet neutral sentiments could not be accurately predicted.

In 2024, *Dervenis, Kanakis & Fitsilis (2024)* conducted both lexicon-based and neural network-based sentiment analyses on student course feedback, finding that the neural network-based method outperformed other methods across all sentiment categories.

Although numerous studies in the literature use course evaluation data for sentiment analysis, there appears to be no research proposing a system that compares traditional machine learning and artificial intelligence tools as a methodology. This study seeks to fill this gap by comparing both traditional machine learning approaches and modern AI methods, ultimately aiming to propose a more effective system for sentiment analysis in course evaluations.

## MATERIALS AND METHODS

This research was carried out using the Design Science Research (DSR) methodology and techniques. In their "Situational Method Engineering" study, *Harmsen, Ernst & Twente (1997)* developed tools aimed at enhancing the flexible applicability of information systems research, offering a framework that directly supports the core elements of the DSR approach. *Venable, Pries-Heje & Baskerville (2017)* examined DSR research methodologies, conducting a review of methodologies suitable for researchers' goals and proposing rules on which DSR methodology should be chosen based on the research objective. Accordingly, the Systems Development Research Methodology (SDRM) was recommended for use in the field of information technology (IT), primarily because the steps in the SDRM methodology focus on determining the foundational elements of an IT system.

In line with the objectives of our study, we opted for SDRM, whose development process consists of five stages (*Nunamaker, Chen & Purdin, 1990*):

(1) Construct a conceptual framework
(2) Develop a system architecture
(3) Analyze and design the system
(4) Build the (prototype) system
(5) Observe and evaluate the system

Within the scope of SDRM, the process first begins by developing a meaningful research question and establishing a conceptual framework that involves analyzing the system's functionalities, requirements, and development processes. Subsequently, an original system architecture is designed according to criteria such as scalability and modularity, and the functions of the components and their relationships are defined. In the third stage, the processes that will support the system's functionalities are designed, alternative solutions are developed, and the most appropriate solution is selected. After these steps, a prototype of the system is created; during this phase, the design and development process is more thoroughly understood, providing insights into possible complexities and issues within the system. In the final stage, the system is observed through case studies or field studies, evaluated using lab or field experiments, and the findings are used to develop new theories or models, thereby deepening the experiences gained throughout the process.

Ethics committee approval was received by Başkent University Academic Evaluation Coordination Office (Ethical approval number: 17162298.600-11).

### Proposed system with SDRM approach

The overall structure of the system is presented in Fig. 1.

### Construct a conceptual framework

In our study, which seeks to address how to improve the consistency and reliability of data in existing CIE processes and how to accelerate these processes, the functionalities and

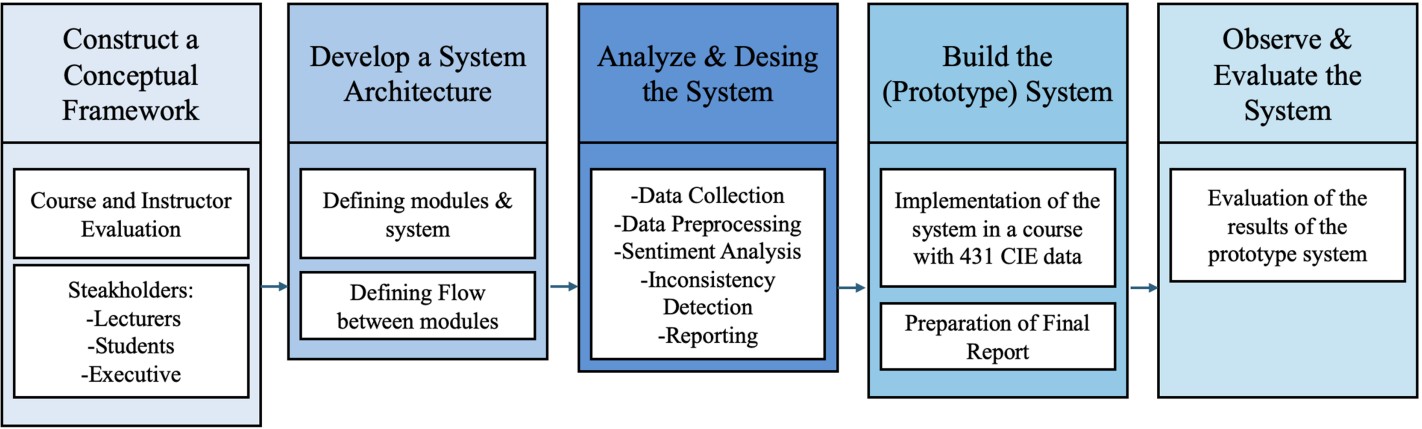

**Figure 1  Structure of the system.**                               

requirements of current systems were examined in detail. Upon closer inspection, it was observed that inconsistencies arise between the responses students provide to structured questions and the opinions they write in the open-ended section in the CIE process, forming a systematic issue. Additionally, it was found that examining the data obtained from the CIE leads to a significant loss of time for administrators.

While developing the conceptual framework to guide the creation of a system intended to propose solutions to these problems, the following considerations were taken into account:

## Develop a system architecture

At this stage, the goal is to design the overall structure of the system to achieve the objectives of the research. The system architecture includes the necessary modules for accurately and consistently processing the data collected from students, as well as the interactions among these modules.

The developed architecture combines core functionalities in a modular structure. The proposed system comprises five modules: data collection, data preprocessing, sentiment analysis, inconsistency detection, and reporting (Fig. 2). Adopting a modular structure is crucial, as each higher education institution may have different CIE processes, and it is therefore important to adapt the system to suit each institution's course evaluation practices.

### Data collection module

This module is responsible for collecting the online CIE forms, which include students' evaluations of the courses they have taken and the instructor who teaches those courses, after each academic semester. During this stage, raw data is generated. This module should be integrated with the student information systems used by higher education institutions.

### Data preprocessing module

This module includes two separate preprocessing steps for both the structured questions and the open-ended section in the CIE. It involves removing elements from the

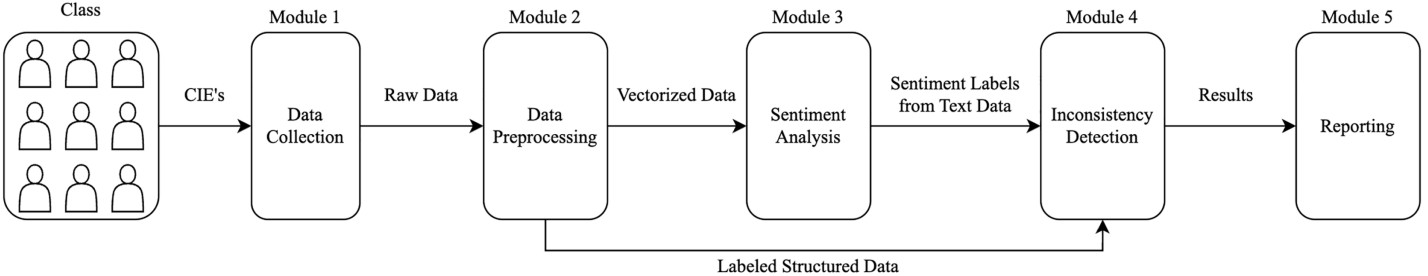

**Figure 2  System architecture.**

open-ended question responses that do not influence the sentiment, as well as manually labeling the structured questions to prepare them for sentiment analysis. Within this module, the PyCharm environment is used with the Python programming language, utilizing libraries such as pandas, re, and nltk.

### Sentiment analysis module

This module involves applying sentiment analysis to the open-ended responses in the CIE, identifying and labeling the emotional state. For sentiment analysis, the 'sklearn' and 'openai' libraries will be employed through the PyCharm environment.

### Inconsistency detection module

This module compares the sentiment label of the open-ended response obtained from the previous module with the label of the structured questions answered by the same student, determining whether the relevant CIE data should be included in the evaluation process.

### Reporting module

This module is responsible for generating reports, based on the consistent evaluations obtained, for presentation to the relevant stakeholders.

## Designing the system

At this stage, the goal is to develop a design that enables the efficient execution of all modules, including data collection, processing, sentiment analysis, inconsistency detection, and reporting. This process involves the planned integration of supervised machine learning algorithms, data preprocessing techniques, and consistency control mechanisms to process the CIE data gathered from students.

## Data collection module design

In this study, the data collection module is based on CIEs obtained online from the student information system of a foundation university (referred to as University A) in Turkey, which has 18,000 (17,908) students. This model can be easily adapted to other higher education institutions that use different student information systems.

At this stage, in order to train the machine learning algorithms to be used in the other modules of the system, all CIEs for the courses offered in the last five years at the aforementioned university were collected *via* the student information system. The

| Table 1 Reliability statistics. | |
|---|---|
| **Reliability statistics** | |
| **Cronbach's Alpha** | **Number of items** |
| 0.992 | 16 |

collected data comprises 16 structured questions (scored on a scale of 1 to 4) and one open-ended question for each CIE. For students, the first 16 questions in the CIE are mandatory, whereas the open-ended question is optional. Since the goal of this study is to detect inconsistencies between the sentiment of the open-ended question response and that of the structured questions, CIEs without responses to the open-ended question were excluded from the dataset. However, if this system is put into practice, the CIEs without an open-ended response should be included, based on the sentiment derived from the structured questions.

Under these conditions, 13.651 CIEs corresponding to 2.874 different courses meeting the specified criteria at University A were identified. The data were stripped of all identifying information and anonymized before being provided to us (University A Ethics Committee Report Document No: 17162298.600-11).

The internal consistency of the instrument was assessed using Cronbach's Alpha coefficient, which yielded a value of 0.992 across 16 items. According to widely accepted benchmarks, a Cronbach's Alpha value above 0.90 indicates excellent reliability. Therefore, the obtained coefficient suggests that the items within the survey demonstrate a very high level of internal consistency and measure the intended construct reliably. This level of reliability supports the use of the instrument for further statistical analyses and strengthens the validity of the findings derived from the data (Table 1).

## Data preprocessing module

Data preprocessing is a crucial step that involves converting raw data into a format suitable for analysis. In this step, both types of data in the dataset (text responses to open-ended questions and quantitative responses to structured questions) are labeled, and any elements that do not affect the sentiment are removed.

Operations carried out in the data preprocessing module (Fig. 3):

1. Text data labeling: The 13,651 CIE records obtained from the data collection module were manually labeled by experts as positive (1), negative (−1), or neutral (0). As a result of this process, 2,914 neutral, 1,955 negative, and 8,782 positive comments were identified and labeled in the dataset.This labeling step will be performed automatically by the chosen algorithm in the sentiment analysis module once the system is implemented. It was done manually here with the aim of determining the most reliable algorithm for sentiment analysis (to be used during the training stage of the algorithm).

2. Numeric data labeling: In the CIE data, 16 structured questions were rated by students on a scale of 1 to 4 (1—Not satisfied at all, 2—Not satisfied, 3—Satisfied, 4—Very

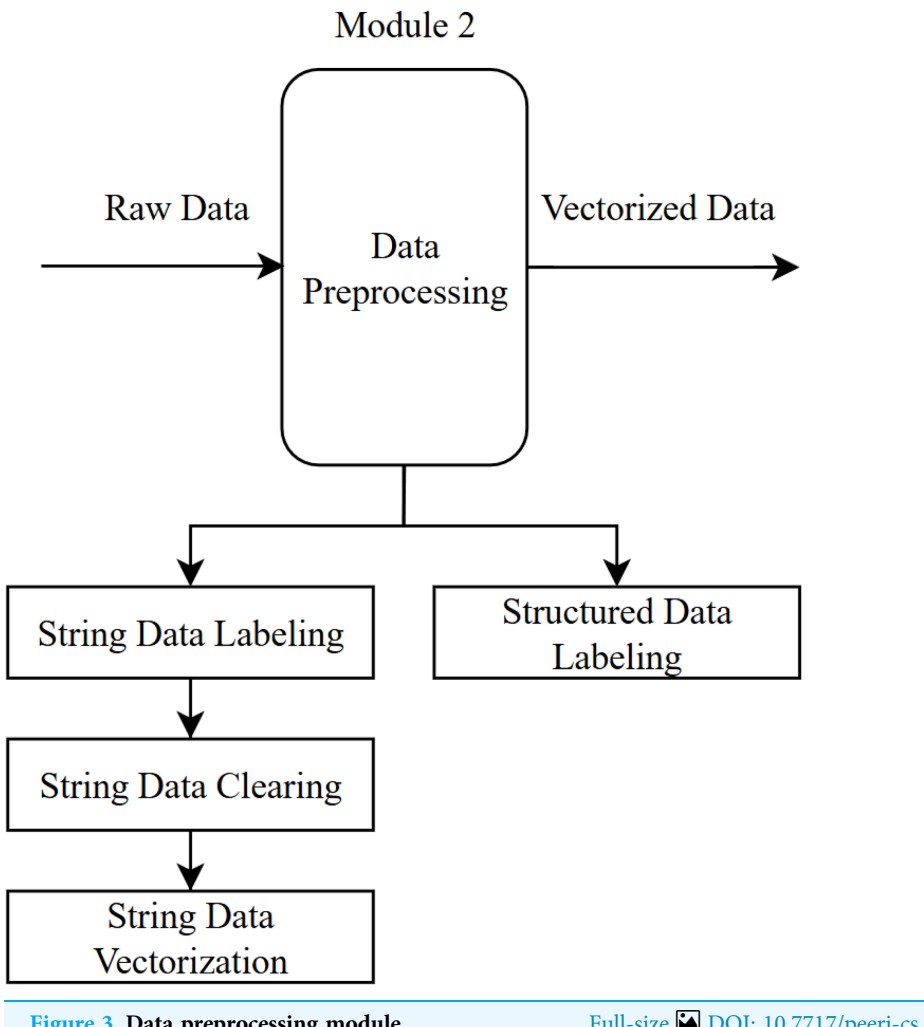

**Figure 3 Data preprocessing module.**

satisfied). In the resulting dataset, an arithmetic mean of the 16 questions was calculated. If this mean (i) was 3 or above, it was labeled as positive (1); (ii) between 2 and 3, neutral (0); and (iii) below 2, negative (−1), indicating the student's overall sentiment for that specific CIE.

3. Text data cleaning: During the data cleaning phase, uppercase letters, punctuation marks, numbers, emojis, and stop words (such as "and," "or," "*etc.*")—which do not affect the sentiment within a sentence—were removed from the dataset.

4. Text data vectorization: The texts were vectorized using the TF-IDF method. TF-IDF is used to determine the importance of a word in a document; frequently occurring words (high term frequency) are considered important within that document. However, if such words also appear commonly in other documents (low inverse document frequency), their importance is reduced, emphasizing the uniqueness of the word. An example of TF-IDF is illustrated in Table 2.

**Table 2 Vectorization example.**

| String data | Vectors |
|---|---|
| "It was a very good lesson, I enjoyed it" | [0.37796447  0.37796446  0.37796447  0.37796447  0.37796448] |
| "It was an inefficient lesson" | [0.57535027  0.57535027  0.57535027  0.57535027] |

## Sentiment analysis module

Sentiment analysis is a field concerned with the computational processing of opinions, emotions, and subjectivity in texts, aiming to develop sentiment-focused information retrieval systems (*Pang & Lee, 2008*). The main objective of this process is to objectively evaluate the emotional expressions in students' open-ended responses. In this module, the machine learning algorithm to be used for sentiment analysis in the system has been selected. Therefore, the preprocessed texts are transferred to supervised machine learning algorithms (support vector machines (SVM), naive Bayes, random forest, decision trees, K-nearest neighbors (KNN), and GPT-4-Turbo Preview), and each text is labeled as positive, negative, or neutral by the respective algorithm.

SVM attempts to find a hyperplane that separates classes by transforming the data into a high-dimensional space. This hyperplane maximizes the minimum distance to the data points closest to the boundary between classes. For non-linearly separable cases, kernel functions are used to project the data into higher dimensions (*Cortes & Vapnik, 1995*).

Naive Bayes operates based on Bayes' theorem and assumes that all features are independent of one another. The probability of belonging to a class is calculated by multiplying the conditional probabilities of each feature. It is commonly used in applications such as text classification and spam filtering (*Maron, 1961*).

Random forest combines multiple decision trees through an ensemble learning method. Each tree is trained on a random subset of the data, and predictions are combined through a voting mechanism. This approach reduces overfitting and ensures high accuracy (*Breiman, 2001*).

Decision trees carry out classification or regression by recursively splitting the data. At each node, the data is divided into two subgroups based on a feature and a threshold value. The final leaves represent the predicted classes or values. The algorithm selects the best split according to criteria like information gain or entropy (*Quinlan, 1986*).

KNN measures the distance of a data point to its K-nearest neighbors for classification or regression. A new data point is classified based on the majority of these neighbors. KNN is a non-parametric method that makes comparisons directly with the data rather than undergoing a training phase (*Cover & Hart, 1967*).

GPT-4 Turbo Preview is a large language model based on OpenAI's Transformer architecture, designed to provide a faster and more cost-effective solution for natural language processing tasks. The model first undergoes pre-training on a broad collection of text to learn linguistic structures, followed by fine-tuning to optimize for user-specific tasks. "Turbo" optimizations offer lower computational costs and faster response times compared to GPT-4 (*OpenAI, 2023*).

Supervised machine learning algorithms are preferred for sentiment analysis in Turkish texts due to their ability to learn from labeled datasets and accurately classify texts in context (*Tokcaer, 2021*; *Akgül, Ertano & Banu, 2016*; *Tuzcu, 2020*; *İlhan & Sağaltıcı, 2020*).

Using the sklearn library, data split into an 80:20 ratio was employed to train the machine learning algorithms mentioned above, targeting high accuracy. It is a common practice to allocate 80% of the dataset for training and 20% for testing when evaluating the performance of machine learning models (*Zilyas & Yılmaz, 2023*; *Almosawi & Mahmood, 2022*; *Ramteke et al., 2016*; *Putra et al., 2022*). During training, the algorithms parameters were kept at default values, and accuracy rates were examined after training.

## Accuracy

Accuracy is a metric that measures how correct a model's predictions are across the entire dataset. It represents the ratio of correctly classified examples to the total number of examples in the dataset and takes into account both true positive (TP) and true negative (TN) predictions.

$$Accuracy = \frac{Number\ of\ Correct\ Predictions}{Total\ Number\ of\ Data}$$

## Mean average precision

Precision is the ratio of truly positive examples among those predicted as positive by the model. Macro average precision calculates precision values for each class separately and then takes the arithmetic mean of those values.

$$Precision = \frac{True\ Positive}{True\ Positive + False\ Positive}$$

$$Macro\ Avg\ Precision = \frac{1}{N} \sum_{i=1}^{N} Precision(i)$$

## Macro average recall

Recall measures how accurately the model identifies truly positive examples as positive. Macro average recall calculates recall values for each class separately and then computes their average.

$$Recall = \frac{True\ Positive}{True\ Positive + False\ Negative}$$

$$Macro\ Avg\ Recall = \frac{1}{N} \sum_{i=1}^{N} Recall(i)$$

Table 3 presents a comparison of the sentiment analysis performance of the algorithms in terms of accuracy, precision, and recall. The results show that the "GPT-4-turbo-preview" model provided by OpenAI achieved the highest performance across all criteria. With 85% accuracy, 88% precision, and 95% recall, this model outperformed the other algorithms. Hence, the GPT-4-Turbo preview algorithm was chosen to be used in the proposed system. The performance comparison across different algorithms reveals that

**Table 3 Sentiment analysis result with machine learning algorithms.**

| Algorithms | Support vector machines (%) | Naive Bayes (%) | Random forest (%) | Decision trees (%) | K-nearest neighbors (%) | GPT-4-Turbo-preview (%) |
|---|---|---|---|---|---|---|
| Accuracy | 83 | 79 | 81 | 76 | 68 | 85 |
| Mean Avg Precision | 76 | 73 | 75 | 67 | 69 | 88 |
| Mean Avg Recall | 76 | 68 | 72 | 66 | 58 | 95 |

**Table 4 Inconsistency detection.**

| String data | Sentiment analysis label | Average numeric data | Numeric data label | Status of data |
|---|---|---|---|---|
| "Our lesson was very productive" | Positive | 1.4 | Negative | Inconsistent |
| "The use of materials in the course was very poor" | Negative | 1 | Negative | Consistent data |

GPT-4 Turbo preview outperforms all traditional machine learning models in terms of accuracy (85%), mean average precision (88%), and mean average recall (95%). Among the classical algorithms, SVM demonstrate the most balanced performance, achieving relatively high scores across all metrics (83% accuracy, 76% precision, and 76% recall). Naive Bayes and random forest follow closely, though with slightly lower overall effectiveness. Decision trees and KNN show the weakest performance, particularly in recall, where KNN achieves only 58%. These results suggest that while traditional models offer competitive baselines, large language models like GPT-4 provide a notable advantage, particularly in capturing nuanced patterns relevant to precision and recall.

### Inconsistency detection module

This module examines the consistency of each CIE labeled in previous steps. Potential inconsistencies are identified by comparing the sentiment label of the textual data with that of the corresponding numeric data (Table 4). For instance, when a student provides a low numeric score but expresses positive sentiments in the open-ended section (or vice versa), this situation is deemed inconsistent.

### Final report module

The final report consolidates the results of all analysis and detection processes to present an overall assessment of course and instructor performance. Its main goal is to provide concrete and reliable information to enhance the accuracy and transparency of course evaluation processes in higher education institutions. The report includes positive, negative, and neutral sentiment trends that summarize student perceptions of the course or instructor. Moreover, by visualizing and evaluating the results obtained after removing inconsistent data, it contributes to strategic decision-making processes for administrators and instructors.
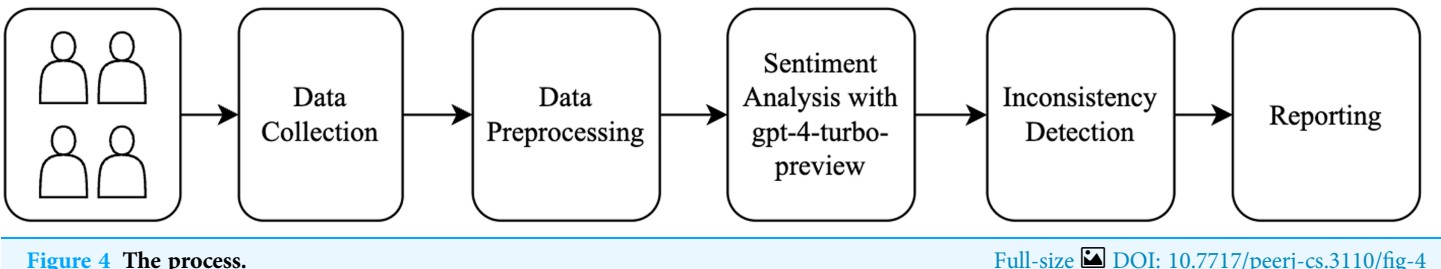

**Figure 4** The process.  

Depending on the institution's needs or the authorization level of the individual viewing the report (administrators, course instructors, *etc.*), the specific information included in the report may vary. However, the final report designed for a course in the proposed system contains the following information (Attachment 1):

- Pie chart displaying the sentiment distribution of consistent feedback on the course
- The average numeric scores for the course
- The highest and lowest average scores given in evaluations
- The percentage of all comments that are consistent/inconsistent

## Build the (prototype) system

At this stage, in order to test the applicability of the developed model in a real course evaluation scenario, a prototype system was created. The prototype system was applied to the CIEs of a course containing 431 comments and survey data. This process is illustrated in Fig. 4.

In the prototype system, the GPT-4-Turbo Preview model, which achieved the highest accuracy rate for sentiment analysis, was used. No changes were made to the operations carried out by the system's other modules.

The dataset employed in this study comprises 431 CIE responses collected within the past academic year from a single course. The sentiment annotation of the qualitative feedback was performed manually by a domain expert with expertise in natural language processing and educational technologies. The majority of the inconsistencies stem from cases where positive quantitative evaluations are accompanied by negative qualitative comments.

As a result of the analyses, out of 431 CIEs, 161 were deemed inconsistent and 270 were deemed consistent. This finding indicates that 37% of the collected CIEs contain misleading results.

A final report was generated using the 270 CIEs labeled as consistent. The following elements are planned to be included in this report.

Providing administrators and instructors with meaningful insights into the strengths and weaknesses of the course, thus helping them make more informed decisions in future course planning and educational processes.

### Observe and evaluate the system

The prototype system was tested on 431 comments and survey data collected for a single course. In this prototype, the sentiment analysis conducted by OpenAI achieved an accuracy rate of 86%.

Upon examining the sentiment labeling results, it was observed that OpenAI generally had difficulty distinguishing whether a comment was positive or neutral. While it performed well in correctly identifying both positive and negative comments, its success rate decreased when differentiating between positive and neutral comments. As seen in the prototype system, removing inconsistent data has led to a final report grounded in more reliable results.

## CONCLUSION AND DISCUSSION

In this study, an innovative system was developed using the SDRM to enhance the reliability and consistency of CIE in higher education. By addressing the systematic inconsistencies between structured and open-ended responses in CIEs, the proposed system has yielded successful results in improving the accuracy and efficiency of evaluation processes.

The system's modular structure, which consists of data collection, preprocessing, sentiment analysis, inconsistency detection, and final reporting modules, offers a solution adaptable to the needs of different higher education institutions. In particular, the sentiment analysis carried out with the GPT-4-Turbo Preview model outperformed other algorithms with an accuracy rate of 85%. While this result confirms the model's robustness in sentiment classification, it also reveals difficulties in distinguishing neutral sentiments.

In prototype system tests, approximately 37% of the 431 collected CIEs were found to be inconsistent. Removing these inconsistencies enabled the creation of a more reliable dataset and allowed the final report to be based on more concrete and accurate information. A high proportion of inconsistent data, such as 37%, can adversely affect the accuracy and reliability of decisions made based on these evaluations. These findings underscore the importance of addressing inconsistencies to enhance the transparency and reliability of CIE-based evaluation processes.

Beyond detecting misleading evaluations, this study also provides administrators and instructors with a comprehensive reporting mechanism on areas requiring improvement. In the future, enhancements might be made to improve the system's ability to better differentiate between neutral and positive sentiments, and to increase its applicability in broader evaluation contexts.

Implementing this system will enable educational institutions to conduct evaluation processes more reliably and accurately, thereby making a significant contribution to decision-making processes.

In future work, we plan to conduct user-centered evaluations by involving department heads and deans to assess the practical utility and effectiveness of the system in real-world decision-making contexts. This pilot study will provide valuable insights into user

satisfaction, system usability, and the actionable value of the generated reports, thereby enabling further refinement to better support administrative and academic stakeholders.

Moreover, we aim to enhance the system's capabilities by incorporating more advanced natural language processing (NLP) techniques to go beyond sentiment analysis and extract meaningful topics from open-ended student feedback. Topic modeling and semantic clustering approaches could provide a richer understanding of recurring themes and specific areas of concern, thereby offering more actionable insights for instructors and administrators. Furthermore, we plan to explore the integration of the system into existing university course evaluation workflows, potentially enabling real-time analysis as evaluations are submitted. This real-time feedback mechanism could facilitate timely interventions and continuous improvement in teaching practices.

## ACKNOWLEDGEMENTS

This research article was produced from a master's thesis written under the supervision of Assoc. Prof. Dr. Erinç Karataş in Ankara University Informatics Program. ChatGPT was used to assist with translating the manuscript into English and refining sentence structure for improved clarity and readability.

### Funding
The authors received no funding for this work.

### Competing Interests
The authors declare that they have no competing interests.

### Author Contributions
- İlker Kocaoğlu conceived and designed the experiments, performed the experiments, analyzed the data, performed the computation work, prepared figures and/or tables, and approved the final draft.
- Erinç Karataş conceived and designed the experiments, authored or reviewed drafts of the article, and approved the final draft.

### Ethics
The following information was supplied relating to ethical approvals (*i.e.*, approving body and any reference numbers):

The University of Baskent granted Ethical approval to carry out the study within its facilities (Ethical Application Ref: 17162298.600-11).

### Data Availability
Raw data and code are available in the Supplemental Files.

## Supplemental Information

Supplemental information for this article can be found online at http://dx.doi.org/10.7717/peerj-cs.3110#supplemental-information.

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
