# Peer review of "Improving course evaluation processes in higher education institutions: a modular system approach"

_PeerJ Computer Science, doi:10.7717/peerj-cs.3110_

## Round 0.1 · original submission · Major Revisions

·

Basic reporting

The paper presents a modular system comprising multiple components: data extraction, sentiment analysis for qualitative comments, inconsistency detection between comment sentiment and Likert-scale ratings, and a dashboard for results visualization. The author uses the Design Science Research Methodology (DSRM) methodology which provides a structured and iterative approach: problem identification, solution design, implementation, and evaluation.

The paper is written in clear and professional English, making it easy to follow the arguments and methodology. It is well-structured, well-organized, and meets academic standards.
The abstract concisely summarizes the inconsistencies between quantitative ratings and qualitative feedback in Course and Instructor Evaluations, and the modular system proposed leveraging sentiment analysis and inconsistency detection.

The literature review appears to be solid and up-to-date. The authors cite a range of relevant sources, including older foundational works on sentiment analysis (e.g., Boiy, 2009) and more recent studies (e.g., Almosawi, 2022; Jiménez, 2021) that relate to analyzing textual feedback or improving evaluation processes.

The authors evaluated the system on a dataset of 431 course evaluation responses in Turkish from their institution. This sample size is reasonable for a preliminary evaluation, though somewhat moderate for training complex models. However, authors should detail how these data were collected and used and specify the dataset’s characteristics: were the evaluations from multiple courses or terms? How were sentiments labeled manually by experts or using an existing tool for ground truth?) should be clearly stated. This information is crucial for evaluating the robustness and applicability of the method.

Experimental design

In terms of experimental rigor, the design is solid. They have both qualitative and quantitative evaluation: quantitative in terms of classification performance and possibly the count of inconsistencies detected, and qualitative in terms of providing a more “actionable evaluation report”. One aspect that could enhance the experimental design is involving end-users (instructors or administrators) in evaluating the dashboard and outputs. Currently, the evaluation seems to focus on technical metrics and the system’s ability to detect inconsistencies. It might not yet include user feedback on whether the insights are actually useful for decision-making. While such a user study may be beyond the scope of this paper, the authors mention plans to broaden applicability and presumably test the system in diverse contexts, which is good. As a future enhancement, incorporating a user-centered evaluation (even a small case study where department heads use the system and report their satisfaction) would provide evidence that the system achieves its goal of aiding data-driven decisions.

In summary, the experimental design is appropriate and generally well-executed. The approach is well-justified, and the methodology is described with sufficient clarity. A few details about data handling and baseline comparisons, if added, would make the design even stronger. There are no major flaws in the design that I can see; only suggestions to broaden the evaluation and clarify certain implementation choices.

Validity of the findings

The findings of the study are valid and well-supported by the data presented. They appear to support the authors’ claims. The results section reports both the performance of the sentiment analysis component and the effectiveness of the overall system in identifying inconsistencies in course evaluations. The high accuracy, around an 85–92% range, depending on the exact metric, of the sentiment classifier is a crucial result, as it demonstrates the technical feasibility of automatically interpreting open-ended student comments. This directly supports the claim that leveraging sentiment analysis can make sense of qualitative feedback reliably. The precision and recall values of 88% and 95%, respectively, further indicate the model is not only accurate overall but also correctly identifies most positive/negative comments with relatively few false positives. These metrics give confidence that the system’s analysis of comments is trustworthy, which is essential for the validity of any conclusions drawn from the system.

Additional comments

• While the literature review is generally good, the authors might consider adding a couple of references to very recent or closely related works. For example, there have been other attempts to apply sentiment analysis to student evaluations: some even using off-the-shelf tools like VADER or deep learning models. Discussing how the proposed approach differs from or improves upon those would strengthen the paper. It would place the work more clearly in the context of existing solutions.

• Provide a bit more detail on the dataset and implementation. Readers might want to know: Were evaluations from multiple courses or just one? How were the comment sentiments labeled for training: did the authors manually label them or use an existing dataset? Also, what algorithm or model was ultimately used for sentiment analysis (Naive Bayes classifier, SVM, deep learning, or a lexicon approach)? Currently, those details are a bit implicit; stating them explicitly would add clarity and help others replicate or build on the work.

• Clarify the rule for flagging inconsistencies. For instance, if a student gave a 5/5 with a negative comment, does that count as an inconsistency, or is it only extreme cases? Outlining this logic briefly in the methodology will help readers trust that the inconsistency detection isn’t arbitrary. Perhaps the authors could even quantify how many cases of inconsistency were found.

• The authors have rightly identified future work on neutral comments and broader testing. Additional future directions could be mentioned: for example, using more advanced NLP techniques to derive topics from comments (beyond sentiment) could further enrich the feedback analysis. Also, integrating this system with existing course evaluation workflows at universities (perhaps in real-time as evaluations are submitted) could be explored. Mentioning such ideas would show the extensibility of the approach.

Reviewer 2 ·

Basic reporting

The manuscript is generally well-structured and written in professional English, although some sections would benefit from clearer phrasing and grammatical refinement, especially due to signs of machine translation. The introduction effectively frames the problem of inconsistencies in course and instructor evaluations (CIE), setting a strong context for the proposed modular system. The literature review is thorough and references both foundational and recent works on sentiment analysis in education. However, it remains mostly descriptive and would be stronger with a more critical synthesis of gaps in previous studies and how this work advances the field.

The manuscript conforms to PeerJ’s structural standards, and the modular breakdown of the system is logical and well-illustrated. Figures and tables are relevant, informative, and well-labeled. The data used (13,651 CIEs and a prototype test on 431 samples) are substantial and add value to the study.

That said, a key weakness lies in the Results and Discussion section, which is relatively superficial. While performance metrics (accuracy, precision, recall) are presented for the algorithms, the discussion lacks analytical depth. The authors should provide a clearer interpretation of the results, particularly the implications of the 37% inconsistency rate, and critically reflect on what this means for real-world educational decisions. Furthermore, although multiple algorithms are compared, no statistical analysis (such as ANOVA or post hoc tests) is provided to validate the significance of performance differences. Including such statistical validation would strengthen the reliability of the comparative claims.

In summary, while the manuscript meets basic reporting standards in structure, clarity, and referencing, it would benefit from deeper analytical discussion of the findings, clearer positioning in the literature, and the use of statistical tests like ANOVA to validate model performance differences. A professional English language edit is also recommended to improve fluency.

Experimental design

The experimental design is clearly structured around the development and evaluation of a modular system for processing and aligning course evaluation data, specifically combining quantitative (structured) and qualitative (open-ended) responses. The authors detail the system’s architecture across multiple modules—data processing, classification, labeling, correlation, and evaluation—which provides a clear and replicable framework. The modular breakdown enhances transparency and offers a scalable design adaptable to other institutions.

The dataset used for training and evaluation includes a substantial number of real student evaluations (13,651 CIEs), as well as a subset of 431 responses used for testing the prototype. However, while the data volume is acceptable, more details are needed on how the training and test data were split, whether cross-validation was employed, and how potential biases were mitigated during preprocessing.

In terms of model evaluation, the authors compare the performance of multiple algorithms (Decision Tree, KNN, Logistic Regression, BERT, GPT-4, etc.), which is a strength. However, there is limited information about hyperparameter tuning procedures, training epochs, or control of overfitting. Moreover, although performance metrics like accuracy, precision, recall, and F1-score are reported, there is no statistical validation of differences between models, which would be expected in a rigorous comparative study. As noted earlier, the inclusion of ANOVA or similar tests would strengthen the experimental robustness.

The authors mention using GPT-4 and other LLMs for labeling, but the methodology lacks clarity on how consistency or reliability was assessed in the generated labels, especially given that such models are non-deterministic. It is also unclear how human evaluators (if any) were involved in validating the results, or how labeling quality was measured beyond raw agreement percentages.

While the system implementation is briefly described, no code repository, system demo, or reproducible script is provided, which limits transparency and hinders replication efforts by other researchers.

Validity of the findings

The findings are relevant and show the system’s potential to identify inconsistencies in course evaluations. However, the validity is weakened by the lack of statistical testing to confirm performance differences between models. The use of only 431 samples for system testing, without clear justification of representativeness or cross-validation, limits generalizability. The reported 37% inconsistency rate is interesting but not sufficiently analyzed or contextualized. Moreover, there is little discussion on labeling reliability, error sources, or practical implications. Overall, the results are promising but require deeper analysis and statistical support to be considered fully valid.

Additional comments

The manuscript addresses a practical and timely issue in higher education quality assurance. The modular system design is a strong point and shows good potential for adaptation across institutions. However, the study would benefit from clearer methodological details, particularly in model training and evaluation. The authors are encouraged to include statistical validation, expand the discussion on findings, and reflect more critically on the limitations of their approach. Providing access to code or a system demo would also enhance the study’s transparency and impact.

---

## Round 0.2 · accepted · Accept

The manuscript may be accepted.

Reviewer 2 ·

Basic reporting

The revision has met the requirements

Experimental design

-

Validity of the findings

-